# Systematic Review: Urine Biomarker Discovery for Inflammatory Bowel Disease Diagnosis

**DOI:** 10.3390/ijms241210159

**Published:** 2023-06-15

**Authors:** Montse Baldan-Martin, María Chaparro, Javier P. Gisbert

**Affiliations:** Gastroenterology Unit, Hospital Universitario de La Princesa, Instituto de Investigación Sanitaria Princesa (IIS-Princesa), Universidad Autónoma de Madrid, Centro de Investigación Biomédica en Red de Enfermedades Hepáticas y Digestivas (CIBEREHD), 28006 Madrid, Spain; mariachs2005@gmail.com (M.C.); javier.p.gisbert@gmail.com (J.P.G.)

**Keywords:** inflammatory bowel disease, Crohn’s disease, ulcerative colitis, biomarkers, urine, proteomics, metabolomics

## Abstract

Inflammatory bowel diseases (IBDs) are chronic, heterogeneous, and inflammatory conditions mainly affecting the gastrointestinal tract. Currently, endoscopy is the gold standard test for assessing mucosal activity and healing in clinical practice; however, it is a costly, time-consuming, invasive, and uncomfortable procedure for the patients. Therefore, there is an urgent need for sensitive, specific, fast and non-invasive biomarkers for the diagnosis of IBD in medical research. Urine is an excellent biofluid for discovering biomarkers because it is non-invasive to sample. In this review, we aimed to summarize proteomics and metabolomics studies performed in both animal models of IBD and humans that identify urinary biomarkers for IBD diagnosis. Future large-scale multi-omics studies should be conducted in collaboration with clinicians, researchers, and industry to make progress toward the development of sensitive and specific diagnostic biomarkers, thereby making personalized medicine possible.

## 1. Introduction

Inflammatory bowel diseases (IBDs), including Crohn’s disease (CD) and ulcerative colitis (UC), are a group of chronic, heterogeneous, and relapsing inflammatory conditions mainly affecting the gastrointestinal tract. IBD is mainly diagnosed at an early age, before 40 years of life, and exerts a significant negative impact on the quality of life of patients because they are at a stage of life with full personal and professional expectations. According to epidemiologic data, the incidence and prevalence of IBD are increasing with time in both high-income and newly industrialized countries, affecting five million people worldwide [1]. In Spain, the reported incidence of IBD is 16 new cases per 100,000 person-years, 7.5 cases per 100,000 person-years for CD, and 8 cases per 1000 person-years for UC [2], highlighting the strategic importance of IBD for society, especially for healthcare systems.

Currently, endoscopy is the gold standard test for assessing mucosal activity and healing in clinical practice. However, it is a costly, time-consuming, invasive, and uncomfortable procedure for the patients and is therefore not the ideal technique to repeat and routinely apply in clinical practice. Therefore, the identification of non-invasive biomarkers with potential clinical use for the diagnosis and prognosis of IBD is urgently needed.

In this context, urine has several advantages over other biological samples, including its very simple and non-invasive collection, its low dynamic range of proteins as it contains a lower abundance of proteins than blood, and the high stability of the sample, which allows reproducible determinations [3,4]. Unlike blood, urine is not subject to homeostatic mechanisms, so it shows greater fluctuations and better reflects changes in the body [5]. Moreover, although the concentration of compounds in urine differs depending on fluid intake, this variation can be compensated for through the use of a set of urinary “housekeeping” peptides as a reference to normalize these fluctuations. In turn, urine reflects not only renal disease but also changes in other organs of the body, such as the intestine [6,7,8]. Urinary biomarkers have shown potential for their use in the diagnosis of different non-urogenital pathologies such as Parkinson’s disease [9], Alzheimer’s disease [10], ovarian, breast and pancreatic cancer [11], osteoarthritis [12], and diabetes [13], among others.

Advances in high-throughput technologies and the availability of large data sets have contributed significantly to the development of omics approaches during the last decade. This multidisciplinary technology, which includes proteomics and metabolomics, has become increasingly important for biomarker research. As many diseases are heterogeneous, biomarkers with ideal sensitivities and specificities are difficult to find. Thus, the combinatorial power of multiple biomarkers could be a potential solution to achieve increased accuracy compared to individual biomarkers. 

In the current review, we will present an update of both preclinical and clinical studies on urinary proteins and metabolites with potential as IBD biomarkers. 

## 2. Methods

### 2.1. Literature Search Strategy

A bibliographic search was designed to identify proteomics and metabolomics studies performed in both animal models of IBD and humans that identified urinary biomarkers for IBD diagnosis. An electronic search was performed in the PubMed and Embase databases up to October 2022, using the following algorithm: (“biomarker” OR “proteins” OR “metabolites”) AND (“proteomics” OR “metabolomics” OR “LC-MS” OR “mass spectrometry” OR “GC-MS” OR “NMR”) AND (“urine” OR “urinary”) AND (“inflammatory bowel disease” OR “ulcerative colitis” OR “Crohn´s disease”). Figure 1 shows the flow chart of reviewed and included studies, following PRISMA 2020 [14].

### 2.2. Proteomics and Metabolomics Approaches to Biomarker Discovery

Recently, omics technologies have emerged as promising tools for revealing molecular pathways via the quantification of differentially expressed molecules associated with different pathological conditions and for identifying biomarkers of IBD [15,16,17,18,19]. The term omics encompasses many technologies, including genomics, metagenomics, transcriptomics, proteomics, and metabolomics. Next-generation omics technologies have allowed for a much more detailed understanding of host and functional genetics in relation to disease [20,21]. It is noteworthy that experimental design is critical in generating accurate and actionable results. Cohort studies are important in understanding human disease; in these studies, various sample types such as stool, plasma, serum, urine, and intestinal tissue can be analyzed using omics pipelines.

Single omics approaches have provided important progress in the understanding of a range of complex diseases. Individually, however, these tools are often unable to capture the true biological complexity of most diseases [21]. Advances in systems biology have enabled the integration of multiple types of omics data, termed multi-omics, which allows for a more comprehensive analysis and may provide important advances in the understanding of disease [22]. The orthogonal information could help to find the causal chain of molecular events, which would not be possible with a single approach [23]. This is especially true in the case of diseases, such as IBD, that arise as a result of the interplay between a range of host and environmental factors [24]. Thus, integrative approaches are appropriate particularly when studying these kinds of diseases as they provide a more holistic view. The advent of this more recent integrative approach represents an exciting possibility for the future of disease characterization, diagnostics, and the prediction of therapeutic response [25,26,27,28].

#### 2.2.1. Metabolomics

Metabolomics technology emerged as a new approach to identifying altered metabolites within all biological systems at an early stage of disease. These metabolites are the ultimate products of gene transcription, translation, and protein function and provide information on metabolic status in human diseases [29,30]. Metabolomics is a promising tool for the discovery of biomarkers and furthering understanding [31]. The typical workflow of metabolomics includes biological sample processing and analysis in terms of total metabolites, which can belong to a specific chemical class depending on the designed approach and the methodologies selected for sample preparation and pre-concentration.

Currently, there are two main analytical platforms for metabolomics studies: nuclear magnetic resonance spectroscopy (NMR) and mass spectrometry (MS), and strategies to perform metabolomics experiments can be divided into untargeted and targeted analyses. In the former, the analyses of the metabolic profiles of organisms are unbiased, comprehensive, and systematic. Targeted metabolomics is the study and analysis of specific metabolites, such as those involved in a particular metabolic pathway or those that are the direct product of drug administration. In targeted analysis, the metabolites under investigation are usually known, and the preparation of samples can be adjusted to reduce the effects of interference from associated metabolites.

Finally, it is important to note that there is no single analytical platform able to perform the complete quantification and identification of all molecules within a sample. Thus, complementary techniques can be used for different analytes depending on the experimental objective and the type of sample.

#### 2.2.2. Proteomics

Proteomics is another area of the omics sciences that is focused on the large-scale study of proteins found in living cells, tissues, or organisms, including their isoforms, post-translational modifications, and their interactions [32,33]. The unique characteristics of the human proteome, which include a highly dynamic range of protein expression, alternative splicing, the interconnectivity of proteins, signaling networks, post-translational modifications, and sequence variations, make such an analysis very challenging [34,35]. The process leading to the clinical implementation of a novel proteomic biomarker is mainly divided into three stages: discovery, verification, and validation, which can be very different and can follow different routes depending on the methods and technologies available [36,37].

Similar to metabolomics, proteomics can be classified into untargeted and targeted approaches. The former aims for the identification and quantification of all detectable proteins in a sample, while targeted proteomics is used to quantify a set of known proteins. MS is the gold standard used for untargeted biomarker discovery studies, with high levels of sensitivity and specificity, while directed and targeted approaches such as antibody-based immunoassays and MS-based assays are the most suitable methods for quantitative biomarker validation and clinical applications [38].

The application and integration of omics techniques in population screening is a promising tool for the detection of early metabolic and protein alterations before disease symptoms appear, which is a basis of a future personalized medicine [39]. The result of these approaches is a panel of differentially expressed proteins or alterations in metabolites that are potential biomarkers.

## 3. Results and Discussion

### 3.1. Urinary Biomarkers for IBD Diagnosis

Urine is considered one of the most valuable biofluids for discovery of disease biomarkers because of its non-invasive method of collection. However, many confounding factors unrelated to disease pathogenesis such as age, sex, diet, and hormonal status create variability in the normal urinary proteome and metabolome. There are two ways to solve the problem: reductionist and holistic big data strategies. The goal of reductionist strategies is to limit the factors to a minimum, establish the direct relation between the factor and its effect, and then validate it in clinical samples. In this approach, animal models should be used since urine is affected by many different factors at any time. Moreover, the entire disease process, including the very early stages, is observable without the interference of any treatment. On the other hand, in the holistic approach, researchers analyze a large number of clinical samples and apply big data analysis to identify the relationships between those independent factors [40].

In this review, a total of 27 research articles from our search remained after a manual inspection of the articles focused on urinary proteomics and metabolomics studies for IBD diagnosis. These studies are discussed below. First, the studies based on animal models of IBD are addressed, followed by the studies based on IBD patients. Table 1 and Table 2 summarize the characteristics and main findings of existing publications based on preclinical studies using animal models and clinical studies, respectively.

### 3.2. Animal Models of IBD

IBD mouse models are an invaluable tool for preclinical research since they provide insights into the complex mechanisms operative in the development and pathogenesis of these diseases. In the last decade, a large number of animal models were developed [41,42,43,44]. These experimental models of IBD have greatly advanced our knowledge about the pathogenesis of acute and chronic intestinal inflammation and have contributed to the development of important therapeutic interventions [45,46]. The animal models most frequently used in the search for urinary biomarkers for IBD diagnosis using metabolomics and proteomics approaches are detailed below.

#### 3.2.1. Chemically Induced Mouse Models

The most widely used preclinical IBD mouse models are those with a disease induced via treatment with chemicals such as dextran sulfate sodium (DSS), trinitrobenzene sulfonic acid (TNBS), and oxazolone, among others.

##### DSS-Induced Colitis Model

In this model, mouse strains are provided cycles of DSS in their drinking water, causing inflammation in the colon which is characterized by ulcers and granulocyte infiltration.

Dong et al. investigated the urine changes in DSS-induced acute colitis by employing an NMR-based metabolomics approach with complementary information on serum clinical chemistry and histopathology. The analysis showed that the DSS-treated mice contained higher concentrations of citric acid cycle intermediates (citrate, 2-oxoglutarate, and fumarate), butyrate, and phenylacetate but lower concentrations of 2-(4-hydroxyphenyl) propanoic acid, methylamine, phenylacetylglycine, indoxyl sulfate, hippurate, 4-cresol glucuronide, 4-cresol sulfate, adipate, azelate, N-methylnicotinate, and methyguanidine than the control mice. Moreover, they found that DSS-induced acute colitis resulted in a depletion of gut microbial cometabolite levels in urine, in addition to an increase in citric acid cycle intermediates. The authors suggest that IBD in this animal model causes a disturbance in the lipid and energy metabolism and damage to the colon and liver, promotes antioxidative and anti-inflammatory responses, and perturbs gut microbiota communities [47].

Schicho et al. performed quantitative profiling of metabolic compounds to detect IBD biomarkers [48]. They characterized 69 urine metabolites using NMR spectroscopy and a targeted analysis to differentiate between DSS-treated mice and healthy animals. Maximal increases in ketone bodies, hypoxanthine, and tryptophan and decreased levels of antioxidant metabolites were observed in urine from DSS colitis mice. The metabolomics patterns generated from the hierarchical multivariate orthogonal partial least-squares (OPLS) data differentiated the DSS-treated mice from the control mice with a predictive power (area under the curve (AUC) = 0.71). The urine samples revealed changes mostly in bacterial metabolites and the metabolites associated with oxidative stress. Nevertheless, the AUC can be assumed to be low considering both the number of metabolites to be analyzed and the laboratory mice used.

##### TNBS-Induced Colitis Model

In the TNBS-induced colitis model, the administration of TNBS results in a preclinical mouse model replicating clinical CD. Upon binding to host proteins, TNBS induces a Th1 immune response characterized by the infiltration of CD4+ T cells, neutrophils, and macrophages. Moreover, transversely spreading inflammation develops, resulting in transmural colitis [49,50].

Qin et al. used quantitative proteomics methods for the discovery of urine biomarkers of irritable bowel disease in a TNBS-induced colitis rat model [51]. In this study, 77 urinary proteins were significantly changed in rats with colitis compared with controls. Among the nine proteins validated via parallel reaction monitoring, carbonic anhydrase 1, neutrophil collagenase, and neutrophil gelatinase-associated lipocalin were previously reported as IBD-associated proteins (all exhibiting trends consistent with those observed in this study), whereas the others (collectrin, beta-mannosidase, sodium-dependent neutral amino acid transporter B(0)AT3, glyceraldehyde-3-phosphate dehydrogenase, and ribonuclease pancreatic gamma-type) were newly discovered. They concluded that urine can be a good source of IBD biomarkers.

Utilizing a metabolomics approach, Zhang et al. characterized the metabolomics profiles of the urine samples of rats with TNBS-induced acute colitis and identified two tryptophan metabolites (4-(2-aminophenyl)-2,4-dioxobutanoic acid and 4,6-cihydroxyquinoline), two gut microbial metabolites (phenyl-acetylglycine and p-cresol glucuronide), and the bile acid 12α-hydroxy-3-oxocholadienic acid in urine [52]. These metabolites, measured via ultra-performance liquid chromatography coupled with electrospray ionization quadrupole time-of-flight mass spectrometry, were associated with damage to the intestinal barrier function, microbiota homeostasis, immune modulation, and the inflammatory response and played important roles in the pathogenesis of IBD.

#### 3.2.2. Genetically Engineered Mouse Models of IBD

##### IL-10-Gene-Deficient Mouse Model

Many models harbor targeted modifications in susceptibility genes identified in human IBD. These models of IBD spontaneously develop colitis and/or ileitis [53]. The most well-known genetically engineered model is the IL-10 knockout mouse. IL-10 is a well-known regulatory cytokine and represents a key IBD (both UC and CD) susceptibility gene. Mice with a targeted disruption of the IL-10 gene suffer from spontaneous pancolitis within several months after birth. Specifically, they present colonic inflammation, characterized by an inflammatory infiltration of lymphocytes, macrophages, and neutrophils. The genetic backgrounds of these mice are modifiers of disease activity, as the penetrance and severity of colitis is markedly different between individual mouse strains, e.g., BALB/c mice develop a more severe disease than C57BL/6 mice.
ijms-24-10159-t001_Table 1Table 1Metabolomics and proteomics studies of biomarker discovery using animal models of IBD.ReferenceAnimal ModelUrinary BiomarkerDetection MethodsSample Size (Discovery Phase)Biological FindingsDong et al. [47]DSS-induced acute UCMetabolitesNMR spectroscopy12 DSS-induced acute UC and 12 controlsHigher levels of citric acid cycle intermediates, butyrate, and phenylacetate and lower concentrations of 2-(4-hydroxyphenyl) propanoic acid, methylamine, phenylacetylglycine, indoxyl sulfate, hippurate, 4-cresol glucuronide, 4-cresol sulfate, adipate, azelate, N methylnicotinate, and methyguanidine in urine samples from DSS-treated mice.Schicho et al. [48]DSS-induced acute UCMetabolitesNMR spectroscopy11 DSS-induced acute UC and 5 controlsIncreased levels of ketone bodies, hypoxanthine, and tryptophan and decreased levels of antioxidant metabolites.Qin et al. [51] TNBS-induced colitis rat modelProteinsTMT-label quantitative LC-MS/MS18 TNBS-induced colitis and 10 controlsA total of 77 proteins were significantly expressed in colitis rats. Nine proteins were validated via PRM. Of these, carbonic anhydrase 1, neutrophil collagenase, and neutrophil gelatinase-associated lipocalin were previously reported as IBD-associated proteins, whereas collectrin, beta-mannosidase, sodium-dependent neutral amino acid transporter B(0)AT3, glyceraldehyde-3-phosphate dehydrogenase and ribonuclease pancreatic gamma-type were newly discovered.Zhang et al. [52]TNBS-induced acute colitisMetabolitesUPLC-ESI-QTOF-MS7 TNBS-induced acute colitis and 10 control ratsTwo tryptophan metabolites (4-(2-aminophenyl)-2,4-dioxobutanoic acid and 4,6-cihydroxyquinoline), two gut microbial metabolites (phenyl-acetylglycine and p-cresol glucuronide), and the bile acid 12a-hydroxy-3-oxocholadienic acid were identified in urine from the TNBS-induced acute colitis rats.Tso et al. [54]IL-10-gene-deficient miceMetabolitesNMR spectroscopy10 IL-10-gene-deficient mice and 10 wild-type miceMetabolomics profile from IL-10-gene-deficient mice is gender-, age-, and disease-specific. Changes in profile patterns that appear to be imperative for the development of intestinal inflammation were observed during the development of IBD.Lin et al. [55]IL-10-gene-deficient miceMetabolitesGC-MS15 IL-10-gene-deficient mice and 10 wild-type miceTryptophan metabolism, fucosylation and fatty acid metabolism were perturbed in IL10−/−mice.Lin et al. [56]IL-10-gene-deficient miceMetabolitesGC-MS20 IL-10-gene-deficient mice and 20 wild-type miceIdentification of fifteen metabolite differences associated with intestinal inflammation and metabolite differences unrelated to inflammation that may indicate novel functions of IL10.Otter et al. [57]IL-10-gene-deficient miceMetabolitesShort-column LC-MSIL-10-gene-deficient mice and wild-type miceThree differential metabolites were associated with colon inflammation. Murdoch et al. [58]IL-10-gene-deficient miceMetabolitesNMR spectroscopy4 IL-10-gene-deficient mice and 4 wild-type miceThe metabolic profiles of control and IL-10-gene-deficient mice diverged substantially with the onset of IBD.DSS-induced acute UC: dextran-sulfate-sodium-induced acute ulcerative colitis; NMR: nuclear magnetic resonance; TNBS-induced colitis rat model: trinitrobenzene-sulfonic-acid- induced colitis rat model; PRM: parallel reaction monitoring; IBD: inflammatory bowel disease; TMT: tandem mass tags; LC-MS/MS: liquid chromatography coupled with tandem mass spectrometry; UPLC-ESI-QTOF-MS: ultra-performance liquid chromatography coupled with electrospray ionization quadrupole time-of-flight mass spectrometry; GC-MS: gas chromatography–mass spectrometry.


Tso et al. explored the urinary metabolome in an IL-10-gene-deficient mouse model of IBD to study the difference in the metabolomics profiles of males and females [54]. Urine samples obtained from mice at the ages of 4, 6, 8, 12, 16, and 20 weeks were analyzed via NMR spectroscopy. A multivariate analysis was employed to assess differences in the metabolomics profiles associated with IBD development and severity (at week 20). The comparison of the gender-differentiating metabolomics profiles between IL-10-gene-deficient mice and wild-type mice during the development of IBD allowed for the identification of changes in profile patterns that appear to be imperative for the development of intestinal inflammation. The main conclusions of the study highlight that the metabolomics profiles in this mouse model of IBD are gender-, age-, and disease-specific, which may have potential clinical implications for the design of biomarkers of disease.

Using gas chromatography–mass spectrometry (GC-MS), Lin et al. demonstrated differences between the urinary metabolite profiling of IL-10-gene-deficient mice and wild type mice [55]. They were able to identify five key metabolic differences, indicating that tryptophan metabolism, fucosylation, and fatty acid metabolism were perturbed in IL-10-gene-deficient mice. Moreover, the perturbation of these pathways could be biochemically related to the intestinal inflammation induced by the absence of IL10. Fucose and xanthurenic acid could be useful as markers of intestinal inflammation. This suggests that non-targeted GC-MS metabolite profiling of IL-10-gene-deficient mice can provide insights into the metabolic effects of IL-10 deficiency and identify potential markers of intestinal inflammation. After that, they identified specific metabolites associated with intestinal inflammation [56]. Fifteen metabolites were associated with intestinal inflammation. Of these metabolites, increased levels of xanthurenic acid were attributed to the increased production of kynurenine metabolites that may induce T-cell tolerance toward intestinal microbiota. On the other hand, eleven metabolites were unaffected by the severity of inflammation. These findings provide new insights into functions of IL 10 unrelated to inflammation.

Otter et al. compared the metabolite profiling of urine from IL-10-gene-deficient mice and wild-type mice using short-column liquid chromatography–mass spectrometry (LC-MS) [57]. They identified two metabolites associated with the degree of inflammation (xanthurenic acid and α-CEHC glucuronide) that could be useful biomarkers of colon inflammation. Nonetheless, previous studies found a convergence of approximately 17% for metabolites differentiated between IBD and controls in human and animal studies.

Using NMR, another study demonstrated urinary metabolic differences in IL-10-gene-deficient mice as the disease developed over time [58]. In this work, a principal component analysis and a partial least-squares-discriminant analysis of urine derived from control and IL-10-gene-deficient mice revealed that while both groups initially had similar metabolic profiles, they diverged substantially after week 8, in which the IL-10-gene-deficient mice showed severe histological injury. The most differences were found in metabolites produced or modulated by gut microflora, including trimethylamine (TMA), concomitant with the known timeline for the development of severe histological injury. This study illustrates that metabolomics is effective at identifying specific urinary metabolites that change with intestinal disease progression in an animal model of IBD.

### 3.3. Urine Samples in IBD

Urine is an abundant body fluid that could be more useful than intestinal tissue and blood for the non-invasive bedside diagnosis, prognosis, and monitoring of IBD. Moreover, the processing of urine samples before analysis is minimal. These advantages have ensured the widespread use of urine as an analytical tool in clinical practice [59]. In this section, we will describe clinical studies focused on biomarker discovery for IBD diagnosis in human urine samples.
ijms-24-10159-t002_Table 2Table 2Metabolomics and proteomics studies of inflammatory bowel disease diagnostics using human urine samples.ReferenceUrinary BiomarkerDetection MethodsSample Size (Discovery Phase)TreatmentsBiological FindingsMaráková et al. [60]ProteinsCE-MS/MS13 IBD and 6 HCAzathioprineDownregulation of serotonin and norepinephrine, and overexpression of histamine and spermidine.Stephens et al. ^[61]^MetabolitesNMR spectroscopy30 CD, 30 UC, and 60 HCMesalamine, corticoesteroid, thiopurine, antimetabolite, and anti-TNFMajor differences between IBD and HC including TCA cycle intermediates, amino acids, and gut microflora metabolites.Schicho et al. [62]MetabolitesNMR spectroscopy20 UC, 20 CD, and 40 HC5-ASA, steroids, 6-MP, azathioprinea, and immunosupresives Higher levels of mannitol, allantoin, xylose, and carnitine and decreased levels of betaine and hippurate.Cracowski et al. [63]MetabolitesGC/electronic impact MS23 CD and 23 HC5-ASA, corticoesteroid, azathioprine, methotrexate, and cyclosporineIncreased iPF2alpha-III concentrations in patients with CD. A correlation was found between urinary iPF2alpha-III and plasma C-reactive protein concentrations.Williams et al. [64] MetabolitesNMR spectroscopy86 CD, 60 UC, and 60 HC5-ASA, azathioprine, and prednisoloneThree specific urinary metabolites related to gut microbial metabolism (hippurate, formate, and 4-cresol sulfate differ between CD, UC, and HC. Martin et al. [65]MetabolitesNMR spectroscopy21 pediatric IBD and 27 HCNo dataThe identification of two readouts of nitrogen metabolic (urea and phenylacetylglutamine) may be relevant to monitoring metabolic status in relation to disease state.Martin et al. [66]MetabolitesLC-MS/MS and GC-MS21 pediatric IBD and 27 HCNo dataMetabolic differences encompass central energy metabolism, amino acids, bile acids, and gut microbial metabolites. Levels of pyroglutamic acid, glutamic acid, glycine, and cysteine were significantly higher in children with IBD.El Hassani et al. [67]MetabolitesGC-MS10 pediatric IBD and 10 HCNoneSignificant difference in VOC profiles between IBD and HC.Yamamoto et al. [68]MetabolitesMSI-CE-MS26 pediatric IBDAnti-TNF, immunomodulator, and 5-ASAIncreased excretions of indoxyl sulfate, hydroxyindoxyl sulfate, phenylacetylglutamine, and sialic acid and lower threonine, serine, kynurenine, and hypoxanthine levels in CD compared to UC patients. Excellent discrimination between groups was achieved based on the urinary serine:indoxylsulfate ratio (AUC = 0.972).Siebert et al. [69]ProteinsCE-MS50 IBD (42 CD and 8 CU), 50 OA, 50 RA, 50 PsA and 50 HC No dataThe classifiers for the five groups demonstrated excellent performances, with AUC values between 0.90 and 0.97 per group.Alonso A et al. [70]MetabolitesNMR spectroscopy1210 IMIDs and 100 HCAnti-TNF, methotrexate, and 5-ASAThe use of the combination of all diagnostic biomarkers as multivariate classifiers demonstrated a good disease prediction accuracy in all IMIDs and particularly in IBD.Keshteli et al. [71]MetabolitesLC-MS/MS and GC-MS53 qUC, 39 IBS and 21 HC5-ASA, immunosuppresants, and biologics.A unique urinary metabolome in IBS patients that could differentiate them from UC patients with an AUC = 0.99.Dawiskiba et al. [72]MetabolitesNMR spectroscopy24 UC, 19 CD and 17 HC5-ASA, azathioprine, and acetaminophenDifferences in citrate, hippurate, trigonelline, taurine, succinate, and 2-hydroxybutyrate levels were found in patients with active IBD. Patients with IBD in remission showed up-regulated levels of acetoacetate and decreased levels of citrate, hippurate, taurine, succinate, glycine, alanine, and formate.Markó et al. [73]ProteinsSE HPLC and MALDI-TOF/MS1 CDMesalamine and azathioprineThe concentration of t-uAlb was found to be 15 times higher than that of their ir-uAlb during the active state. Bjerrum et al. [74]MetabolitesNMR spectroscopy41 aUC, 33 qUC, and- 25 HCMesalazine, glucocorticoids, azathioprine, and anti-TNFThe metabolic profiles of the urine did not allow for differentiation between active UC, inactive UC, and HC.Stanke-Labesque et al. [75]MetabolitesLC-MS/MS32 CD, 28 UC, and 30 HCAzathioprine, 5-ASA, and glucocorticoidsLTE4 urinary excretion was significantly higher in CD and UC patients compared to HC. LTE4 levels were higher in patients with active disease than in patients in remission.Keshteli et al. [76]MetabolitesNMR spectroscopy and LC-MS/MS38 CD5-ASA, azathioprine/6-MP, methotrexate, corticosteroids, and anti-TNFEndoscopic recurrence was associated with an increased concentration of urinary levoglucosan. Rutgeerts score was positively correlated with levoglucosan and propylene glycol levels.Keshteli et al. [77]MetabolitesNMR spectroscopy and LC-MS/MS20 UC5-ASA and immunosuppresantsThree metabolites in urine (trans-aconitate, cysteine and acetamide) and three metabolites in serum (3-hydroxybutyrate, acetoacetate and acetone) were responsible for the discrimination of UC patients with clinical relapse.CE-MS/MS: capillary electrophoresis tandem mass spectrometry; IBD: inflammatory bowel disease; HC: healthy controls; NMR: nuclear magnetic resonance; CD: Crohn´s disease; UC: ulcerative colitis; TCA: tricarboxylic acid; VOC: volatile organic compounds; LC-MS/MS: liquid chromatography coupled with tandem mass spectrometry; GC-MS: gas chromatography–mass spectrometry; MSI-CE-MS: multisegment injection capillary electrophoresis-mass spectrometry; AUC: area under the curve; OA: osteoarthritis; RA: rheumatoid arthritis; PsA: psoriatic arthritis; IMIDs: immune-mediated inflammatory diseases; qUC: quiescent ulcerative colitis; IBS: irritable bowel syndrome; SE HPLC: size exclusion high-performance liquid chromatography; t-uAlb: non-immune-reactive urinary albumin; uAlb: immune-reactive urinary albumin; aUC: active ulcerative colitis; LTE4: leukotriene E(4); 5-ASA: 5-aminosalicylic acid; 6-MP: mercaptopurine.


#### 3.3.1. Urinary Markers in the Differentiation of IBD Patients from Healthy Controls (HCs)

Maráková et al. determined the levels of 12 biogenic amines (histamine, serotonin, dopamine, norepinephrine, epinephrine, putrescine, cadaverine, spermine, spermidine, tyramine, tryptamine, and phenylethylamine) in human urine as potential IBD biomarkers [60]. By using capillary electrophoresis coupled with MS, they were able to identify decreases in serotonin and norepinephrine and increases in histamine and spermidine in 13 CD patients treated with azathioprine when compared with 6 controls. The authors suggested these findings could be used to clarify the mechanism of IBD. However, this study has several limitations, for example, levels of these biogenic amines were not detected in the subjects. Moreover, the patients analyzed were treated with azathioprine and therefore could not be assessed for biomarkers with potential diagnostic uses. Finally, the small sample size must be considered, and the different parameters that can cause concentration changes in circulating and urinary-eliminated biogenic amines (diet, consumption of alcoholic beverages, smoking habit, physical activity, stress, co-morbidity, and co-medications) must also be considered.

Stephens et al. examined urine samples via NMR spectroscopy in combination with targeted techniques to identify metabolomics profiles able to distinguish 60 IBD patients from 60 HC patients [61]. Tricarboxylic acid cycle intermediates, amino acids, and gut microflora metabolites showed clear differences between the study groups. Regarding IBD types, no differences in urine metabolites were seen between CD and UC. In addition, drug and dietary therapies, as well as surgical resections, were confounding factors that interfered with the understanding of the metabolic changes associated with these diseases. This work highlights the potential of metabolomics to distinguish IBD patients from HC patients, but it shows that careful consideration must be given to establishing disease-representative cohorts free of confounding factors. For this purpose, urine samples from patients before any drug therapies are the ideal samples for potentially diagnosing CD and UC at the outset.

Schicho et al. characterized 71 urine metabolites from patients with 40 patients with IBD patients and 40 HCs via NMR spectroscopy and a targeted analysis [62]. They used multi-block principal component analysis and hierarchical orthogonal projections to latent structures discriminant analysis to examine differences in the metabolites. Patients with IBD showed increased levels of mannitol, allantoin, xylose, and carnitine and decreased levels of betaine and hippurate. The metabolomics profiling on urine discriminated between HC and IBD patients. On the contrary, the metabolic differences between the CD and UC cohorts were less pronounced. The main limitation of the study is that the patients were taking medications including 5-aminosalicylate drugs, azathioprine, and corticosteroids. Therefore, further studies with newly diagnosed treatment-naïve patients are needed.

Cracowski et al. used GC/electronic-impact MS to measure concentrations of urinary isoprostaglandin F2alpha type III in 23 CD patients compared to 23 HC to test whether lipid peroxidation correlates to clinical relapse and inflammation [63]. The analysis showed increased concentrations of urinary isoprostaglandin F2alpha type III in patients with CD. Moreover, a significant correlation was found between urinary isoprostaglandin F2alpha type III and plasma C-reactive protein concentrations, suggesting a link between lipid peroxidation and inflammation. The authors concluded that the isoprostaglandin F2alpha type III quantification must be investigated as a prognosis biomarker in patients suffering from CD.

A metabolomic analysis using NMR was performed by Williams et al. to identify specific urinary metabolites related to gut bacteria that differ between CD patients (n = 86), UC (n = 60) and HC (n = 60) [64]. Hippurate levels were differentially expressed between the three cohorts, whose lowest levels were found in CD (*p* < 0.0001). Moreover, this group of patients showed higher levels of formate and lower levels of 4-cresol sulfate compared to UC patients or HC (*p* = 0.0005 and *p* = 0.0002, respectively). This multivariate analysis was able to distinguish these three cohorts.

The first work in children was carried out by Martin et al. in 2016 [65]. In this study, the metabolic profiles of 21 pediatric patients with IBD and 27 healthy children were analyzed via NMR spectroscopy. Metabolic differences included central energy metabolism, amino acid, and gut microbial metabolic pathways. The analysis showed two readouts of nitrogen metabolism (urinary urea and phenylacetylglutamine) that could be relevant to monitor metabolic status in the course of disease. In 2017, they identified the metabolite profiles of urine samples from 21 IBD patients by applying LC and GC coupled with targeted MS; the patients were monitored three times over one year (at baseline and at 6 and 12 months) [66]. The levels of pyroglutamic acid, glutamic acid, glycine, and cysteine were significantly higher in children with IBD over the course of the study. These findings suggest that glutathione cannot be optimally synthesized and replenished. Moreover, non-invasive urinary bile acid profiling can assess altered hepatic and intestinal barrier dysfunctions. Nevertheless, this metabolite signature should be further investigated in future studies with larger sample sizes. 

Another study conducted by El Hassani et al. performed a differential metabolomics analysis of urinary volatile organic compounds (VOCs) in de novo pediatric IBD patients to evaluate their accuracy as alternative and non-invasive diagnostic biomarkers for IBD [67]. A total of five UC, five CD, and ten HC urine samples were analyzed using GC-MS. These results showed a significant difference in urinary VOC profiles between IBD patients and HCs (AUC = 0.78). The authors suggested that the analysis of urinary VOCs has potential as a means of identifying non-invasive biomarkers for pediatric IBD diagnoses. Nevertheless, the sample size recruited in this study was very small. Therefore, future large clinical studies are needed to corroborate these results.

Yamamoto et al. investigated the metabolomes of urine samples obtained from 18 pediatric CD patients and 8 UC patients, aiming at differentiating these IBD subtypes [68]. The authors used multisegment injection-capillary electrophoresis-MS. A statistical analysis revealed higher urinary excretions of indoxyl sulfate, hydroxyindoxyl sulfate, phenylacetylglutamine, and sialic acid in CD patients when compared to UC patients but lower excretions of threonine, serine, kynurenine, and hypoxanthine. An excellent discrimination of CD from UC was achieved based on the urinary serine/indoxylsulfate ratio (AUC = 0.972). The authors suggested that this determination may complement or replace existing strategies in the diagnosis and early management of children with IBD. Although these findings are interesting, the major limitation of this single-center retrospective pilot study was the modest and unbalanced sample size without an HC group. 

#### 3.3.2. Comparison of Proteomics and Metabolomics Profiles between Patients with IBD and Patients with Other Inflammatory Diseases

Siebert et al. used urinary proteome analysis to identify specific biomarkers for different inflammatory conditions (distinct forms of arthritis (rheumatoid arthritis, psoriatic arthritis, and osteoarthritis) or chronic inflammatory conditions) and HCs, using 50 patients per group [69]. They found classifiers for the five study groups that demonstrated excellent discrimination performances between the groups, with AUC values between 0.90 and 0.97 per group. These data suggest that different inflammatory conditions could provide unique peptide profiles of this kind. No validation in independent cohorts has been performed since this study.

Using NMR, Alonso et al. analyzed the urine metabolomes in a discovery cohort of 1210 patients with immune-mediated inflammatory diseases and 100 HCs to identify new metabolite biomarkers of diagnosis and disease activity [70]. The six most prevalent immune-mediated inflammatory diseases were analyzed: rheumatoid arthritis, psoriatic arthritis, psoriasis, systemic lupus erythematosus, CD, and UC. Metabolite association with disease diagnosis and activity was analyzed using multivariate linear regression. After multiple test corrections, the most significantly associated metabolite biomarkers were validated in an independent cohort of 1200 patients and 200 HCs. Multiple comparisons of these IMIDs allowed for the identification of hub metabolites and also the characterization of clinically similar disease clusters based exclusively on urine metabolite profiles. These common molecular findings are in line with the genetic risk in IMIDs identified through genome-wide association studies. Using combinations of all diagnostic biomarkers as multivariate classifiers demonstrated a good disease prediction accuracy in all immune-mediated inflammatory diseases and particularly in IBD. The findings of this study represent an important step in the development of more efficient and less invasive diagnostic and disease-monitoring methods in immune-mediated inflammatory diseases. 

Keshteli et al. analyzed urine metabolites from 53 UC patients in clinical remission, 39 patients with irritable bowel syndrome, and 21 HCs to identify novel pathophysiological targets and biomarkers that could discriminate irritable bowel syndrome from related conditions [71]. The metabolomics analyses were performed using LC-MS/MS and GC-MS. The study revealed that patients with irritable bowel syndrome had a unique urinary metabolome that could differentiate them from UC patients, with an AUC = 0.99. Amino acids and organic acids were the most important metabolites for this differentiation. The main limitations of the study were the small number of patients and the differences in age and gender between the three groups of subjects. Moreover, the definition of UC remission was based on partial Mayo scoring, not an endoscopic assessment.

#### 3.3.3. Urinary Markers in Disease Activity Evaluation in IBD Patients

Dawiskiba et al. evaluated the utility of urine metabolomics analysis in diagnosing and monitoring IBD [72]. For this purpose, they compared urine samples from 24 patients with UC, 19 patients with CD, and 17 HCs using NMR spectroscopy. The activity of UC was assessed with the Simple Clinical Colitis Activity Index, while the activity of CD was determined using the Harvey–Bradshaw Index. They found that the comparison between patients with active IBD and patients in remission provided good partial-least-squares-discriminant analysis models. The urine metabolite levels associated with activities in these models were increased levels of glycine and decreased levels of acetoacetate. Moreover, there were statistically significant differences between the patients with active IBD and the HCs. The urine metabolites showing differences in this comparative study were citrate, hippurate, trigonelline, taurine, succinate, and 2-hydroxybutyrate. On the other hand, the comparison between patients with IBD in remission and the HCs showed up-regulated urine levels of acetoacetate and decreased levels of citrate, hippurate, taurine, succinate, glycine, alanine, and formate.

To identify disease activity in CD, Markó et al. measured immune-reactive urinary albumin concentrations via immune-turbidimetry and non-immune-reactive urinary albumin via size-exclusion high performance liquid chromatography [73]. The albumin peak was collected and assessed via reversed-phase high-performance liquid chromatography and gel electrophoresis. During the active state of the disease, they found a concentration of non-immune-reactive urinary albumin that was 15 times higher than that of immune-reactive urinary albumin. Matrix-assisted laser desorption ionization–time of flight mass spectrometry measurements identified α1-acid-glycoprotein and Zn-α2-glycoprotein as major proteins and albumin as a minor protein. The authors concluded that α1-acid-glycoprotein and/or Zn-α2-glycoprotein could be ideal biomarkers of CD disease activity. It is noteworthy that only one patient was included in this study. Moreover, further studies are needed to evaluate the practicability of these new potential biomarkers in clinical settings.

Another study carried out by Bjerrum et al. compared urine from 41 patients with active UC, 33 with quiescent UC, and 25 HCs via NMR spectroscopy to define new potential biomarkers [74]. The analysis showed that the urine metabolomics profiles were identical in all three groups of patients.

Stanke-Labesque et al. used LC-MS/MS to analyze the urinary excretion of leukotriene E4 in 32 CD patients, 28 UC patients, and 30 HCs. Increased LTE4 urinary excretion was observed in CD and UC patients compared to the HCs. Moreover, these levels were higher in patients with active disease than in patients in remission (*p* < 0.001), who showed leukotrieneE4 levels similar to those of the HCs. The authors concluded that cysteinyl leukotriene pathway activation could contribute to the inflammation and that the increased urinary excretion of leukotriene E4 may be an interesting non-invasive biomarker of disease activity in IBD patients [75]. However, future experimental and clinical studies are required to confirm these findings.

#### 3.3.4. Urinary Metabolites as Biomarkers for Clinical Relapse in Patients with CD and UC

Keshteli et al. using high-resolution NMR spectroscopy and LC-MS/MS on urinary samples from 38 CD patients, identified an association between an increased concentration of levoglucan and endoscopic recurrence [76]. Moreover, the Rutgeert score was positively correlated with levoglucosan and propylene glycol levels. These findings could contribute to improving our understanding of the pathophysiological mechanisms of CD recurrence after ileocolonic resection and the identification of urinary biomarkers related to the recurrence of CD and its severity. However, further studies with larger sample sizes and comprehensively assessed clinical and dietary factors will be necessary to validate these results.

Another work carried out by Keshteli et al. studied metabolomics factors associated with risk of UC clinical relapse within 12 months [77]. They identified three metabolites in urine (trans-aconitate, cysteine, and acetamide) and three metabolites in serum (3-hydroxybutyrate, acetoacetate, and acetone) that were able to differentiate UC patients with clinical relapse. The major limitation of the study was the small sample size (12 UC remission patients and 7 UC relapse patients). Therefore, further well-designed prospective studies investigating these parameters with larger numbers of patients should be performed.

### 3.4. Correlation between Animal Models of IBD and Human IBD Diagnostic Markers

We summarize the main metabolites identified in both animal and human studies. Hypoxanthine was increased in a DSS-induced acute colitis model and decreased in CD patients compared to UC patients [48,68]. Levels of 4-cresol sulfate were decreased in a DSS-induced acute colitis model [47] and patients with CD [64]. Additionally, the metabolite hippurate was observed to be decreased in a DSS-induced acute colitis model and in patients with IBD [47,62,64,72]. Conflicting results were found for citrate, with increased levels found in in DSS-induced acute colitis models and downregulated expression identified in IBD patients compared to controls [47,72]. These data showed a low degree of similarity between animal models of IBD and human studies, making it difficult to correlate the results from different animal models with IBD patients.

## 4. Conclusions

Novel non-invasive biomarkers for IBD diagnosis with higher sensitivities and specificities that can be easily implemented in clinical practice are needed. Significant technological advances over the past decade have allowed for the systematic, holistic, and unbiased characterization of alterations in proteins and metabolites in urine samples associated with the identification of novel biomarkers and disease conditions, making these approaches more promising. Urine can be non-invasively sampled in abundance, making it the ideal liquid biopsy resource to identify biomarkers. Several studies have reported proteins and metabolites as biomarker candidates for IBD diagnosis. In our opinion, the most promising biomarkers for CD and UC diagnosis could be the metabolites hippurate and formate, which have been identified via NMR [64], while leukotriene E4 and levoglucan may be potential biomarkers of disease activity and clinical relapse, respectively [75,76]. Nevertheless, there are numerous limitations and challenges. First, there is insufficient evidence thus far to support the use of urinary biomarkers in clinical practice for IBD diagnosis owing to a lack of robust validation in large-scale longitudinal studies. The diagnostic cut-off value of the biomarker must be established, and the positive and negative predictive values must be demonstrated in the general population. Secondly, several aspects must be taken into consideration when designing studies for urinary IBD biomarker discovery. One of them is the quality control of human samples because urinary proteins and metabolites are easily affected by the sampling time, medication use, diet, age, gender and level of physical activity. Moreover, urine samples from subjects with proteinuria and lipiduria should be excluded because such conditions could indicate renal diseases [78]. Regarding the time of urine collection, the first urine in the morning is preferred to minimize variability from food metabolism. Several studies have tested for the consistency of urine protein and metabolite determinations after repeated cycles of freezing and thawing or sampling at different times [79,80]. Moreover, a standard protocol for the processes used to collect, handle and store specimens should be established prior to analysis to reduce the inconsistency of results between studies. In addition, a single molecule could not be a strong biomarker for clinical practice. A combination of multimodal biomarkers using an integrative approach that simultaneously incorporates multiple metabolites and proteins from urine as well as other biofluids, such as serum or plasma, might considerably improve the diagnosis of IBD. Another limitation identified in this review is that proteomics studies are fewer and less in-depth than metabolomics results, indicating the need to fill this gap. In summary, proteomics and metabolomics technologies have great potential for discovering novel biomarkers for IBD diagnosis, which could offer a promising alternative to invasive endoscopies. Nevertheless, gaps between discovery research and clinical applications remain to be filled. The usage of -omics in medicine for diagnostics would be a major shift in the paradigms in medicine (which currently typically focus on individual markers). Future large-scale multi-omics studies should be conducted in collaboration with clinicians, researchers, and industry to progress toward the development of sensitive and specific diagnostic biomarkers, thereby making personalized medicine possible.

## Figures and Tables

**Figure 1 ijms-24-10159-f001:**
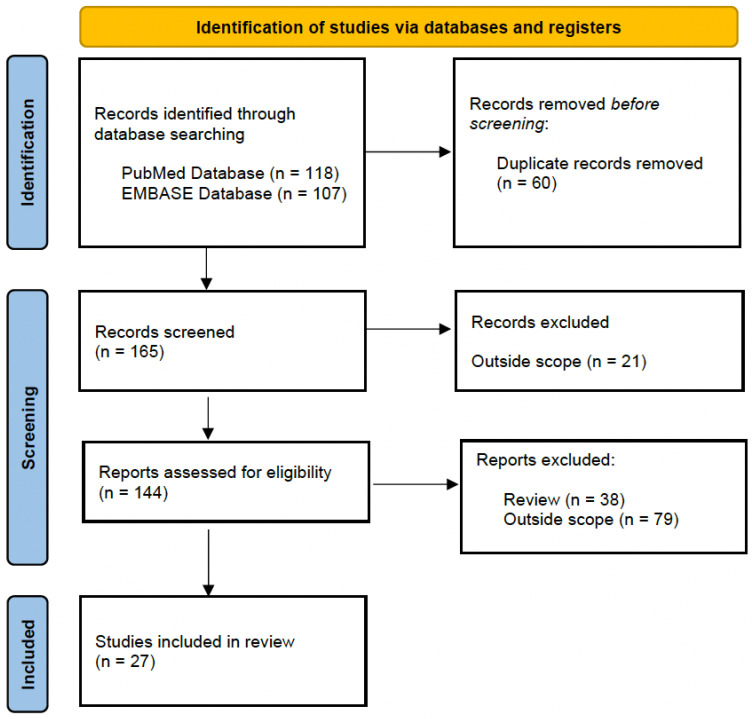
Flow chart detailing selection process of included studies in this review of urine biomarkers discovery for IBD diagnosis, following PRISMA 2020.

## Data Availability

Not applicable.

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
