# Peer review of "Systematic Review: Urine Biomarker Discovery for Inflammatory Bowel Disease Diagnosis"

_ijms, 2023, doi:10.3390/ijms241210159_

Round 1

Reviewer 1 Report

Overall this is an excellent review paper. But there is one critical mistake I want to point out. The idea that "Unlike blood, urine is not subject to homeostatic mechanisms, so it shows greater fluctuations and better reflects changes in the body" is not from the reference 5 whichs is "Urinary proteomics. Clin Chim Acta. 2007;375:49-56". I searched the whole paper and did not find the most important word "homeostatic". This important idea was first proposed in the paper "Urine—an untapped goldmine for biomarker discovery? Sci China Life Sci, 2013, doi: 10.1007/s11427-013-4574-1", which is not correctly referred. Other than that I think this review manuscript is excellent.

Author Response

We appreciate the reviewer´s comments. Following the reviewer´s suggestion, we have included the correct reference in the new version of the manuscript.

Reviewer 2 Report

Interesting trial to find urinary biomarkers  for IBD patients, the animal studies are less convincing, but human observations may be of significance

Author Response

We thank to the reviewer for his comments and we have included a new section in the manuscript entitled “Correlation between animal models of IBD and human IBD diagnostic markers”. In this part, we have detailed the few common metabolites identified in both animal models and human studies.

Reviewer 3 Report

The authors have made a good, sincere attempt to review the literature that exists on using urine as a biomarker for IBD. They attempt to clarify the positives and the shortcomings of the few major studies that were detailed here, especially the human subject studies. 

I think this manuscript will, however, benefit from a better illustration of the facts, and while tabulation is good, the authors should try and bring out their opinion on a situation where urine could be indeed used as a diagnostic tool, and what their recommendations would be.  I will clarify my comments below:

1. Firstly I do not think it is necessary to describe what proteomics and metabolomics is, in the "methods" section. Sections 2.2.1 and 2.2.2 can be abridged and/or removed. The introductory section 2.2 preface is enough according to me.   It was, however, good to include the selection process of the literature/cited articles (Sec 2.1).

2. While it is definitely interesting read of what kind of metabolites and proteins can be detected in animal models, if there is a direct comparison that is made with the human subject studies. Some attempt has been made: "In this work, principal component analysis and partial least-squares-discriminant analysis of urine derived from control and IL-10 gene-deficient mice revealed that while both groups initially had similar metabolic profiles, they diverged substantially with the onset of IBD". How so? a comparison with references could be helpful.

3. As the authors have rightly pointed out the caveats in each of the human subject studies, I would recommend that be added as a separate column in Table 2 with a heading of "treatments or interventions" , or something similar. 

4. Authors could advise on recommendations of which metabolites or proteins could be the most promising, or even rank them based on these literature reviews. How successful, comparitively, could urine be as a biomarker for diagnoisis of inital onset, versus remission versus relapse?

5. Also, as a segue,  mouse IBD model urine metabolites and human subject metabolites could be compared and debated for a more enriched discussion of how these animal IBD models either correlate (or not) with human IBD diagnostic markers.

minor point:

Table 2 should specifically mention "using human urine samples" in the heading

Consistently good quality overall. No comments.

Author Response

Response 1: We have considered your comments and we have abridged the sections 2.2.1 and 2.2.2 in the new version.

Response 2: We agree with the reviewer, and we have rewritten the paragraph describing the results obtained in the study for greater comprehension and clarity.

Response 3: In view of the reviewer comment, we have added an additional column in Table 2 with the treatments.

Response 4: We have added a sentence with our opinion about the most promising metabolites for IBD diagnosis, markers of disease activity and clinical relapse, in the new manuscript (page 13, lines 621-624).

Response 5: We agree with the reviewer and a new section (3.3) was included detailing the common metabolites that have been identified both in animal models and in human studies.

Response: We have included “human” in table tittle 2.

Round 2

Reviewer 1 Report

No further suggestion.